# Modular Networks:
# Learning to Decompose Neural Computation

**Louis Kirsch**[*]
Department of Computer Science
University College London
mail@louiskirsch.com

**Julius Kunze**
Department of Computer Science
University College London
juliuskunze@gmail.com

**David Barber**
Department of Computer Science
University College London
david.barber@ucl.ac.uk

## Abstract

Scaling model capacity has been vital in the success of deep learning. For a typical network, necessary compute resources and training time grow dramatically with model size. Conditional computation is a promising way to increase the number of parameters with a relatively small increase in resources. We propose a training algorithm that flexibly chooses neural modules based on the data to be processed. Both the decomposition and modules are learned end-to-end. In contrast to existing approaches, training does not rely on regularization to enforce diversity in module use. We apply modular networks both to image recognition and language modeling tasks, where we achieve superior performance compared to several baselines. Introspection reveals that modules specialize in interpretable contexts.

## 1 Introduction

When enough data and training time is available, increasing the number of network parameters typically improves prediction accuracy [16, 6, 14, 1]. While the largest artificial neural networks currently only have a few billion parameters [9], the usefulness of much larger scales is suggested by the fact that human brain has evolved to have an estimated 150 trillion synapses [19] under tight energy constraints. In deep learning, typically all parts of a network need to be executed for every data input. Unfortunately, scaling such architectures results in a roughly quadratic explosion in training time as both more iterations are needed and the cost per sample grows. In contrast, usually only few regions of the brain are highly active simultaneously [20]. Furthermore, the modular structure of biological neural connections [28] is hypothesized to optimize energy cost [8, 15], improve adaption to changing environments and mitigate catastrophic forgetting [26].

Inspired by these observations, we propose a novel way of training neural networks by automatically decomposing the functionality needed for solving a given task (or set of tasks) into reusable modules. We treat the choice of module as a latent variable in a probabilistic model and learn both the decomposition and module parameters end-to-end by maximizing a variational lower bound of the likelihood. Existing approaches for conditional computation [25, 2, 21] rely on regularization to avoid a module collapse (the network only uses a few modules repeatedly) that would result in poor

---

[*]now affiliated with IDSIA, The Swiss AI Lab (USI & SUPSI)

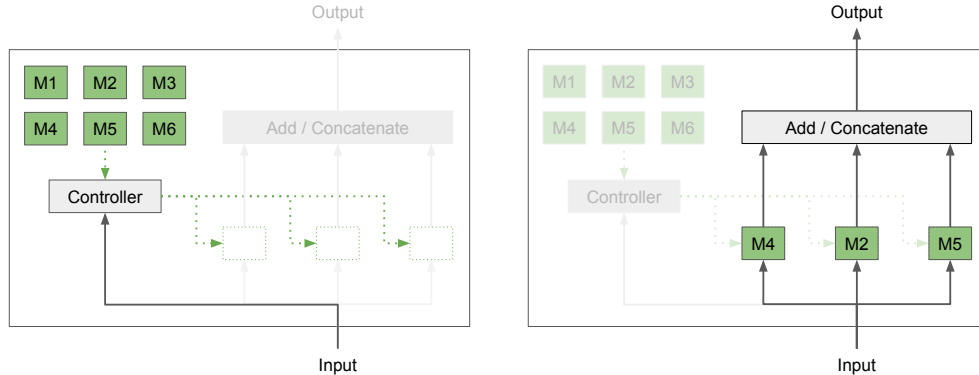

(a) Based on the input, the controller selects $K$ modules from a set of $M$ available modules. In this example, $K = 3$ and $M = 6$.

(b) The selected modules then each process the input, with the results being summed up or concatenated to form the final output of the modular layer.

Figure 1: Architecture of the modular layer. Continuous arrows represent data flow, while dotted arrows represent flow of modules.

performance. In contrast, our algorithm learns to use a variety of modules without such a modification and we show that training is less noisy compared to baselines.

A small fixed number out of a larger set of modules is selected to process a given input, and only the gradients for these modules need to be calculated during backpropagation. Different from approaches based on mixture-of-experts, our method results in fully deterministic module choices enabling low computational costs. Because the pool of available modules is shared between processing stages (or time steps), modules can be used at multiple locations in the network. Therefore, the algorithm can learn to share parameters dynamically depending on the context.

## 2    Modular Networks

The network is composed of functions (modules) that can be combined to solve a given task. Each module $f_i$ for $i \in \{1, \ldots, M\}$ is a function $f_i(x; \theta_i)$ that takes a vector input $x$ and returns a vector output, where $\theta_i$ denotes the parameters of module $i$. A modular layer, as illustrated in Figure 1, determines which modules (based on the input to the layer) are executed. The output of the layer concatenates (or sums) the values of the selected modules. The output of this layer can then be fed into a subsequent modular layer. The layer can be placed anywhere in a neural network.

More fully, each modular layer $l \in \{1, \ldots, L\}$ is defined by the set of $M$ available modules and a controller which determines which $K$ from the $M$ modules will be used. The random variable $a^{(l)}$ denotes the chosen module indices $a^{(l)} \in \{1, \ldots, M\}^K$. The controller distribution of layer $l$, $p(a^{(l)}|x^{(l)}, \phi^{(l)})$ is parameterized by $\phi^{(l)}$ and depends on the input to the layer $x^{(l)}$ (which might be the output of a preceding layer).

While a variety of approaches could be used to calculate the output $y^{(l)}$, we used concatenation and summation in our experiments. In the latter case, we obtain

$$y^{(l)} = \sum_{i=1}^{K} f_{a_i^{(l)}} \left( x^{(l)}; \theta_{a_i^{(l)}} \right) \tag{1}$$

Depending on the architecture, this can then form the input to a subsequent modular layer $l + 1$. The module selections at all layers can be combined to a single joint distribution given by

$$p(a|x, \phi) = \prod_{l=1}^{L} p(a^{(l)}|x^{(l)}, \phi^{(l)}) \tag{2}$$

The entire neural network, conditioned on the composition of modules $a$, can be used for the parameterization of a distribution over the final network output $y \sim p(y|x, a, \theta)$. For example, the

---

**Algorithm 1** Training modular networks with generalized EM

---

Given dataset $D = \{(x_n, y_n) \,|\, n = 1, \dots, N\}$
Initialize $a_n^*$ for all $n = 1, \dots, N$ by sampling uniformly from all possible module compositions
**repeat**
    Sample mini-batch of datapoint indices $I \subseteq \{1, \dots, N\}$         ▷ Partial E-step
    **for each** $n \in I$ **do**
        Sample module compositions $\tilde{A} = \{\tilde{a}_s \sim p(a_n|x_n, \phi) | s = 1, \dots, S\}$
        Update $a_n^*$ to best value out of $\tilde{A} \cup \{a_n^*\}$ according to Equation 11
    **end for**
    **repeat** k times         ▷ Partial M-step
        Sample mini-batch from dataset $B \subseteq D$
        Update $\theta$ and $\phi$ with gradient step according to Equation 8 on mini-batch $B$
**until** convergence

---

final module might define a Gaussian distribution $\mathcal{N}\left(y|\mu, \sigma^2\right)$ as the output of the network whose mean and variance are determined by the final layer module. This defines a joint distribution over output $y$ and module selection $a$

$$p(y, a|x, \theta, \phi) = p(y|x, a, \theta)p(a|x, \phi) \tag{3}$$

Since the selection of modules is stochastic we treat $a$ as a latent variable, giving the marginal output

$$p(y|x, \theta, \phi) = \sum_a p(y|x, a, \theta)p(a|x, \phi) \tag{4}$$

Selecting $K$ modules at each of the $L$ layers means that the number of states of $a$ is $M^{KL}$. For all but a small number of modules and layers, this summation is intractable and approximations are required.

## 2.1 Learning Modular Networks

From a probabilistic modeling perspective the natural training objective is maximum likelihood. Given a collection of input-output training data $(x_n, y_n), n = 1, \dots, N$, we seek to adjust the module parameters $\theta$ and controller parameters $\phi$ to maximize the log likelihood:

$$\mathcal{L}(\theta, \phi) = \sum_{n=1}^{N} \log p(y_n|x_n, \theta, \phi) \tag{5}$$

To address the difficulties in forming the exact summation over the states of $a$ we use generalized Expectation-Maximisation (EM) [17], here written for a single datapoint

$$\log p(y|x, \theta, \phi) \geq -\sum_a q(a) \log q(a) + \sum_a q(a) \log \left(p(y|x, a, \theta)p(a|x, \phi)\right) \equiv L(q, \theta, \phi) \tag{6}$$

where $q(a)$ is a variational distribution used to tighten the lower bound $L$ on the likelihood. We can more compactly write

$$L(q, \theta, \phi) = \mathbb{E}_{q(a)}[\log p(y, a|x, \theta, \phi)] + \mathbb{H}[q] \tag{7}$$

where $\mathbb{H}[q]$ is the entropy of the distribution $q$. We then seek to adjust $q, \theta, \phi$ to maximize $L$. The partial M-step on $(\theta, \phi)$ is defined by taking multiple gradient ascent steps, where the gradient is

$$\nabla_{\theta, \phi} L(q, \theta, \phi) = \nabla_{\theta, \phi} \mathbb{E}_{q(a)}[\log p(y, a|x, \theta, \phi)] \tag{8}$$

In practice we randomly select a mini-batch of datapoints at each iteration. Evaluating this gradient exactly requires a summation over all possible values of $a$. We experimented with different strategies to avoid this and found that the Viterbi EM [17] approach is particularly effective in which $q(a)$ is constrained to the form

$$q(a) = \delta(a, a^*) \tag{9}$$

where $\delta(x, y)$ is the Kronecker delta function which is 1 if $x = y$ and 0 otherwise. A full E-step would now update $a^*$ to

$$a_{\text{new}}^* = \underset{a}{\arg\max}\, p(y|x, a, \theta)p(a|x, \phi) \tag{10}$$

for all datapoints. For tractability we instead make the E-step partial in two ways: Firstly, we choose the best from only $S$ samples $\tilde{a}_s \sim p(a|x, \phi)$ for $s \in \{1, ..., S\}$ or keep the previous $a^*$ if none of these are better (thereby making sure that $L$ does not decrease):

$$a^*_{\text{new}} = \underset{a \in \{\tilde{a}_s | s \in \{1,...,S\}\} \cup \{a^*\}}{\text{argmax}} p(y|x, a, \theta)p(a|x, \phi) \qquad (11)$$

Secondly, we apply this update only for a mini-batch, while keeping the $a^*$ associated with all other datapoints constant.

The overall stochastic generalized EM approach is summarized in Algorithm 1. Intuitively, the algorithm clusters similar inputs, assigning them to the same module. We begin with an arbitrary assignment of modules to each datapoint. In each partial E-step we use the controller $p(a|x, \phi)$ as a guide to reassign modules to each datapoint. Because this controller is a smooth function approximator, similar inputs are assigned to similar modules. In each partial M-step the module parameters $\theta$ are adjusted to learn the functionality required by the respective datapoints assigned to them. Furthermore, by optimizing the parameters $\phi$ we train the controller to predict the current optimal module selection $a^*_n$ for each datapoint.

Figure 2 visualizes the above clustering process for a simple feed-forward neural network composed of 6 modular layers with $K = 1$ modules being selected at each layer out of a possible $M = 3$ modules. The task is image classification, see Section 3.3. Each node in the graph represents a module and each datapoint uses a path of modules starting from layer 1 and ending in layer 6. The width of the edge between two nodes $n_1$ and $n_2$ represents the number of datapoints that use the first module $n_1$ followed by $n_2$; the size of a node represents how many times that module was used. Figure 2 shows how a subset of datapoints starting with a fairly uniform distribution over all paths ends up being clustered to a single common path. The upper and lower graphs correspond to two different subsets of the datapoints. We visualized only two clusters but in general many such clusters (paths) form, each for a different subset of datapoints.

## 2.2 Alternative Training

Related work [25, 3, 21] uses two different training approaches that can also be applied to our modular architecture. REINFORCE [30] maximizes the lower bound

$$B(\theta, \phi) \equiv \sum_a p(a|x, \phi) \log p(y|x, a, \theta) \leq \mathcal{L}(\theta, \phi) \qquad (12)$$

on the log likelihood $\mathcal{L}$. Using the log-trick we obtain the gradients

$$\nabla_\phi B(\theta, \phi) = \mathbb{E}_{p(a|x,\phi)}[\log p(y|x, a, \theta)\nabla_\phi \log p(a|x, \phi)] \qquad (13)$$
$$\nabla_\theta B(\theta, \phi) = \mathbb{E}_{p(a|x,\phi)}[\nabla_\theta \log p(y|x, a, \theta)] \qquad (14)$$

These expectations are then approximated by sampling from $p(a|x, \phi)$. An alternative training algorithm is the noisy top-k mixture of experts [25]. A mixture of experts is the weighted sum of several parameterized functions and therefore also separates functionality into multiple components. A gating network is used to predict the weight for each expert. Noise is added to the output of this gating network before setting all but the maximum $k$ units to $-\infty$, effectively disabling these experts. Only these $k$ modules are then evaluated and gradients backpropagated. We discuss issues with these training techniques in the next section.

## 2.3 Avoiding Module Collapse

Related work [25, 3, 21] suffered from the problem of missing module diversity ("module collapse"), with only a few modules actually realized. This premature selection of only a few modules has often been attributed to a self-reinforcing effect where favored modules are trained more rapidly, further increasing the gap [25]. To counter this effect, previous studies introduced regularizers to encourage different modules being used for different datapoints within a mini-batch. In contrast to these approaches, no regularization is needed in our method. However, to avoid module collapse, we must take sufficient gradient steps within the partial M-step to optimize both the module parameters $\theta$, as well as the controller parameters $\phi$. That is, between each E-step, there are many gradient updates for both $\theta$ and $\phi$. Note that this form of training is critical, not just to prevent module collapse but to

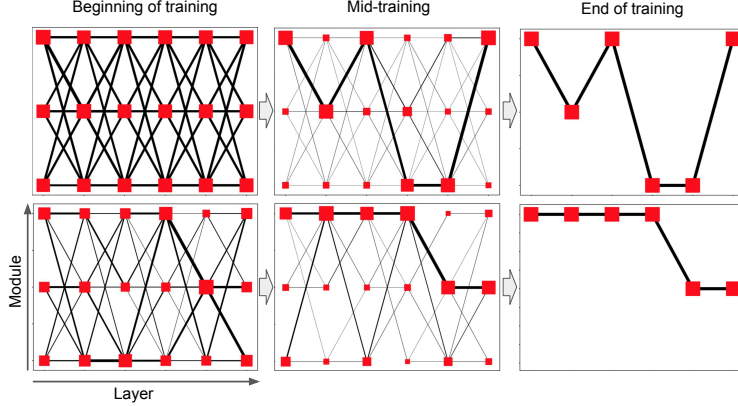

Figure 2: Two different subsets of datapoints (top and bottom) that use the same modules at the end of training (right) start with entirely different modules (left) and slowly cluster together over the course of training (left to right). Nodes in the graph represent modules with their size proportional to the number of datapoints that use this module. Edges between nodes $n_1$ and $n_2$ and their stroke width represent how many datapoints first use module $n_1$ followed by $n_2$.

obtain a high likelihood. When module collapse occurs, the resulting log-likelihood is *lower* than the log-likelihood of the non-module-collapsed trained model. In other words, our approach is not a regularizer that biases the model towards a desired form of a sub-optimal minimum – it is a critical component of the algorithm to ensure finding a high-valued optimum.

## 3 Experiments

To investigate how modules specialize during training, we first consider a simple toy regression problem. We then apply our modular networks to language modeling and image classification. Alternative training methods for our modular networks are noisy top-k gating [25], as well as REINFORCE [3, 21] to which we will compare our approach. Except if noted otherwise, we use a controller consisting of a linear transformation followed by a softmax function for each of the $K$ modules to select. Our modules are either linear transformations or convolutions, followed by a ReLU activation. Additional experimental details are given in the supplementary material.

In order to analyze what kind of modules are being used we define two entropy measures. The module selection entropy is defined as

$$H_a = \frac{1}{BL} \sum_{l=1}^{L} \sum_{n=1}^{B} \mathbb{H}\left[p(a_n^{(l)}|x_n, \phi)\right] \tag{15}$$

where $B$ is the size of the batch. $H_a$ has larger values for more uncertainty for each sample $n$. We would like to minimize $H_a$ (so we have high certainty in the module being selected for a datapoint $x_n$). Secondly, we define the entropy over the entire batch

$$H_b = \frac{1}{L} \sum_{l=1}^{L} \mathbb{H}\left[\frac{1}{B} \sum_{n=1}^{B} p(a_n^{(l)}|x_n, \phi)\right] \tag{16}$$

Module collapse would correspond to a low $H_b$. Ideally, we would like to have a large $H_b$ so that different modules will be used, depending on the input $x_n$.

### 3.1 Toy Regression Problem

We demonstrate the ability of our algorithm to learn conditional execution using the following regression problem: For each data point $(x_n, y_n)$, the input vectors $x_n$ are generated from a mixture of Gaussians with two components with uniform latent mixture probabilities $p(s_n = 1) = p(s_n = 2) = \frac{1}{2}$ according to $x_n|s_n \sim \mathcal{N}(x_n|\mu_{s_n}, \Sigma_{s_n})$. Depending on the component $s_n$, the target $y_n$ is

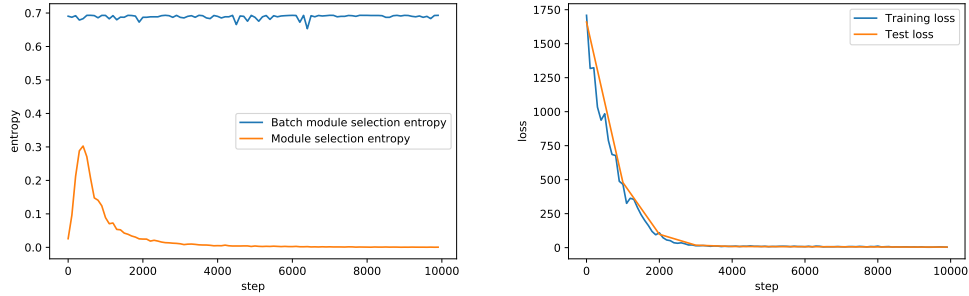

(a) Module composition learned on the toy dataset.     (b) Minimization of loss on the toy dataset.

Figure 3: Performance of one modular layer on toy regression.

generated by linearly transforming the input $x_n$ according to

$$y_n = \begin{cases} Rx_n & \text{if } s_n = 1 \\ Sx_n & \text{otherwise} \end{cases} \tag{17}$$

where $R$ is a randomly generated rotation matrix and $S$ is a diagonal matrix with random scaling factors.

In the case of our toy example, we use a single modular layer, $L = 1$, with a pool of two modules, $M = 2$, where one module is selected per data point, $K = 1$. Loss and module selection entropy quickly converge to zero, while batch module selection entropy stabilizes near $\log 2$ as shown in Figure 3. This implies that the problem is perfectly solved by the architecture in the following way: Each of the two modules specializes on regression of data points from one particular component by learning the corresponding linear transformations $R$ and $S$ respectively and the controller learns to pick the corresponding module for each data point deterministically, thereby effectively predicting the identity of the corresponding generating component. Thus, our modular networks successfully decompose the problem into modules yielding perfect training and generalization performance.

## 3.2 Language Modeling

Modular layers can readily be used to update the state within an RNN. This allows us to model sequence-to-sequence tasks with a single RNN which learns to select modules based on the context. For our experiments, we use a modified Gated Recurrent Unit [5] in which the state update operation is a modular layer. Therefore, $K$ modules are selected and applied at each time step. Full details can be found in the supplement.

We use the Penn Treebank[2] dataset, consisting of 0.9 million words with a vocabulary size of 10,000. The input of the recurrent network for each timestep is a jointly-trained embedding vector of size 32 that is associated with each word.

We compare the EM-based modular networks approach to unregularized REINFORCE (with an exponential moving average control variate) and noisy top-k, as well as a baseline without modularity, that uses the same $K$ modules for all datapoints. This baseline uses the same number of module parameters per datapoint as the modular version. For this experiment, we test four configurations of the network being able to choose $K$ out of $M$ modules at each timestep: 1 out of 5 modules, 3 out of 5, 1 out of 15, and 3 out of 15. We report the test perplexity after 50,000 iterations for the Penn Treebank dataset in Table 1.

When only selecting a single module out of 5 or 15, our modular networks outperform both baselines with 1 or 3 fixed modules. Selecting 3 out of 5 or 15 seems to be harder to learn, currently not outperforming a single chosen module ($K = 1$). Remarkably, apart from the controller network, the baseline with three static modules performs three times the computation and achieves worse test perplexity compared to a single intelligently selected module using our method. Compared to the REINFORCE and noisy-top-k training methods, our approach has lower test perplexities for each module configuration.

Table 1: Test perplexity after 50,000 steps on Penn Treebank

| Type | #modules ($M$) | #parallel modules ($K$) | test perplexity |
|---|---|---|---|
| EM Modular Networks | 15 | 1 | **229.651** |
| EM Modular Networks | 5 | 1 | 236.809 |
| EM Modular Networks | 15 | 3 | 246.493 |
| EM Modular Networks | 5 | 3 | 236.314 |
| REINFORCE | 15 | 1 | 240.760 |
| REINFORCE | 5 | 1 | 240.450 |
| REINFORCE | 15 | 3 | 274.060 |
| REINFORCE | 5 | 3 | 267.585 |
| Noisy Top-k ($k = 4$) | 15 | 1 | 422.636 |
| Noisy Top-k ($k = 4$) | 5 | 1 | 338.275 |
| Baseline | 1 | 1 | 247.408 |
| Baseline | 3 | 3 | 241.294 |

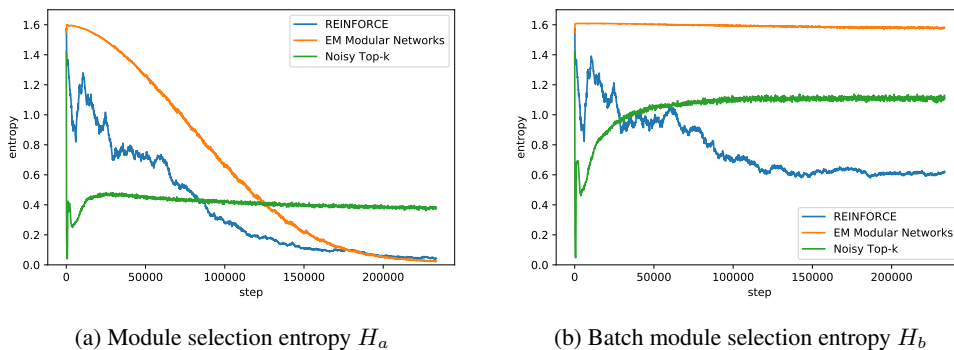

(a) Module selection entropy $H_a$       (b) Batch module selection entropy $H_b$

Figure 4: Modular networks are less noisy during optimization compared to REINFORCE and more deterministic than noisy top-k. Our method uses all modules at the end of training, shown by a large batch module selection entropy. The task is language modeling on the Penn Treebank dataset.

We further inspect training behavior in Figure 4. Using our method, all modules are effectively being used at the end of training, as shown by a large batch module selection entropy in Figure 4b. Additionally, the optimization is generally less noisy compared to the alternative approaches and the method quickly reaches a deterministic module selection. Figure 5 shows how the module selection changes over the course of training for a single batch. At the beginning of training, the controller essentially has no preference over modules for any instance in the batch. Later in training, the selection is deterministic for some datapoints and finally becomes fully deterministic.

For language modeling tasks, modules specialize in certain grammatical and semantic contexts. This is illustrated in Table 2, where we observe specialization on numerals, the beginning of a new

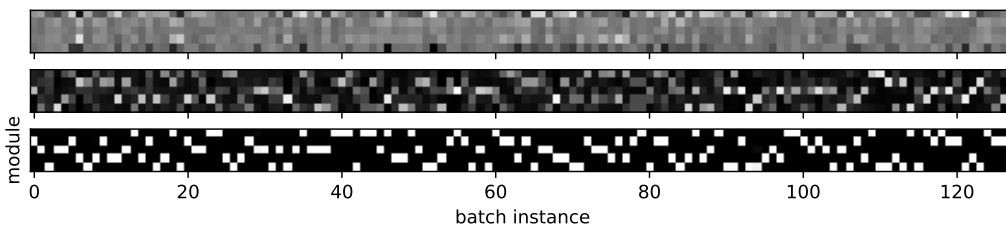

Figure 5: A visualization of the controller distribution for a particular mini-batch, choosing $K = 1$ out of $M = 5$ modules. Training progresses from the top image to the bottom image. A black pixel represents zero probability and a white pixel represents probability 1.

Table 2: For a few out of $M = 15$ modules (with $K = 1$), we show examples of the corresponding input word which they are invoked on (highlighted) together with surrounding words in the sentence.

| Module 1 | Module 3 | Module 14 |
|---|---|---|
| ... than **\<number\>** \<number\> ... | ... Australia **\<new sentence\>** A ... | ... said **the** acquired ... |
| ... be **substantially** less ... | ... opposition **\<new sentence\>** I ... | ... on **the** first ... |
| ... up **\<number\>** \<number\> ... | ... said **\<new sentence\>** But ... | ... that **the** carrier ... |
| ... \<number\> **million** was ... | ... teachers **for** the ... | ... to **the** recent ... |
| ... \$ **\<number\>** billion ... | ... result **\<new sentence\>** That ... | ... and **the** sheets ... |
| ... \<number\> **million** of ... | ... \<new sentence\> **but** the ... | ... and **the** naczelnik ... |
| ... \$ **\<number\>** billion ... | ... based **on** the ... | ... if **the** actual ... |
| ... by **\<number\>** to ... | ... business **\<new sentence\>** He ... | ... say **the** earnings ... |
| ... yield **\<number\>** \<number\> ... | ... rates **\<new sentence\>** This ... | ... in **the** third ... |
| ... debt **from** the ... | ... offer **\<new sentence\>** Federal ... | ... brain **the** skin ... |
| ... | ... | ... |

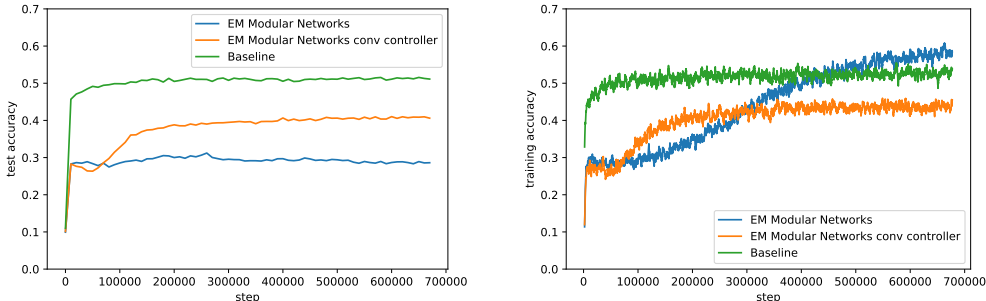

Figure 6: Modular networks test (left) and training accuracy (right) for a linear controller and a convolutional controller compared to the non-modularized baseline.

sentence and the occurrence of the definite article *the*, indicating that the word to be predicted is a noun or adjective.

### 3.3 Image Classification

We applied our method to image classification on CIFAR10 [13] by using a modularized feed-forward network. Compared to [21], we not only modularized the two fully connected layers but also the remaining three convolutional layers. Details can be found in the supplement.

Figure 6 shows how using modules achieves higher training accuracy compared to the non-modularized baseline. However, in comparison to the language modeling tasks, this does not lead to improved generalization. We found that the controller overfits to specific features. In Figure 6 we therefore compared to a more constrained convolutional controller that reduces overfitting considerably. Shazeer et al. [25] make a similar claim in their study and therefore only train on very large language modeling datasets. More investigation is needed to understand how to take advantage of modularization in tasks with limited data.

## 4 Related Work

Learning modules and their composition is closely related to mixtures of experts, dating back to [11, 12]. A mixture of experts is the weighted sum of several parameterized functions and therefore also separates functionality into multiple components. Our work is different in two major aspects. Firstly, our training algorithm is designed such that the selection of modules becomes fully deterministic instead of a mixture. This enables efficient prediction such that only the single most likely module has to be evaluated for each of the $K$ distributions. Secondly, instead of having a single selection of modules, we compose modules hierarchically in an arbitrary manner, both sequentially and in parallel. The latter idea has been, in part, pursued by [10], relying on stacked mixtures of experts instead of a single selection mechanism. Due to their training by backpropagation of entire mixtures,

summing over all paths, no clear computational benefits have yet been achieved through such a form of modularization.

Different approaches for limiting the number of evaluations of experts are stochastic estimation of gradients through REINFORCE [30] or noisy top-k gating [4]. Nevertheless, both the mixture of experts in [3] based on REINFORCE as well as the approach by [25] based on noisy top-k gating require regularization to ensure diversity of experts for different inputs. If regularization is not applied, only very few experts are actually used. In contrast, our modular networks use a different training algorithm, generalized Viterbi EM, enabling the training of modules without any artificial regularization. This has the advantage of not forcing the optimization to reach a potentially sub-optimal log-likelihood based on regularizing the training objective.

Our architecture differs from [25] in that we don't assign a probability to every of the $M$ modules and pick the $K$ most likely but instead we assign a probability to each composition of modules. In terms of recurrent networks, in [25] a mixture-of-experts layer is sandwiched between multiple recurrent neural networks. However, to the best of our knowledge, we are the first to introduce a method where each modular layer is updating the state itself.

The concept of learning modules has been further extended to multi-task learning with the introduction of routing networks [21]. Multiple tasks are learned jointly by conditioning the module selection on the current task and/or datapoint. While conditioning on the task through the use of the multi-agent Weighted Policy Learner shows promising results, they reported that a single agent conditioned on the task and the datapoint fails to use more than one or two modules. This is consistent with previous observations [3, 25] that a RL-based training without regularization tends to use only few modules. We built on this work by introducing a training method that no longer requires this regularization. As future work we will apply our approach in the context of multi-task learning.

There is also a wide range of literature in robotics that uses modularity to learn robot skills more efficiently by reusing functionality shared between tasks [22, 7]. However, the decomposition into modules and their reuse has to be specified manually, whereas our approach offers the ability to learn both the decomposition and modules automatically. In future work we intend to apply our approach to parameterizing policies in terms of the composition of simpler policy modules.

Conditional computation can also be achieved through activation sparsity or winner-take-all mechanisms [27, 23, 24] but is hard to parallelize on modern accelerators such as GPUs. A solution that works with these accelerators is learning structured sparsity [18, 29] but often requires non-sparse computation during training or is not conditional.

## 5   Conclusion

We introduced a novel method to decompose neural computation into modules, learning both the decomposition as well as the modules jointly. Compared to previous work, our method produces fully deterministic module choices instead of mixtures, does not require any regularization to make use of all modules, and results in less noisy training. Modular layers can be readily incorporated into any neural architecture. We introduced the modular gated recurrent unit, a modified GRU that enables minimalistic sequence-to-sequence models based on modular computation. We applied our method in language modeling and image classification, showing how to learn modules for these different tasks.

Training modular networks has long been a sought-after goal in neural computation since this opens up the possibility to significantly increase the power of neural networks without an increase in parameter explosion. We have introduced a simple and effective way to learn such networks, opening up a range of possibilities for their future application in areas such as transfer learning, reinforcement learning and lifelong learning. Future work may also explore how modular networks scale to larger problems, architectures, and different domains. A library to use modular layers in TensorFlow can be found at `http://louiskirsch.com/libmodular`.

### Acknowledgments

We thank Ilya Feige, Hippolyt Ritter, Tianlin Xu, Raza Habib, Alex Mansbridge, Roberto Fierimonte, and our anonymous reviewers for their feedback. This work was supported by the Alan Turing Institute under the EPSRC grant EP/N510129/1. Furthermore, we thank IDSIA (The Swiss AI Lab)

for the opportunity to finalize the camera ready version on their premises, partially funded by the ERC Advanced Grant (no: 742870).

## Footnotes

[2] http://www.fit.vutbr.cz/~imikolov/rnnlm/simple-examples.tgz

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
