[Supplementary Material · Modular_Networks_NeurIPS_18_supplementary.pdf]

## A Modular GRU

We propose the *modular GRU*, defined by

$$z_t = \sigma(W_z \cdot [h_{t-1}, x_t])$$
$$r_t = \sigma(W_r \cdot [h_{t-1}, x_t])$$
$$\tilde{h}_t = \text{ReLu}(\sum_{i=0}^{K} f_{a_i^{(t)}}([r_t \odot h_{t-1}, x_t]))$$
$$h_t = (1 - z_t) \odot h_{t-1} + z_t \odot \tilde{h}_t$$

where $h_t$ is the RNN's hidden state at time $t$, $z_t$ and $r_t$ are gate values and $\tilde{h}_t$ is the candidate state produced by the modular layer at time step $t$. Both the controller $p(a_t|h_{t-1}, x_t, \phi)$ and the modules $f_m([r_t \odot h_{t-1}, x_t]; \theta_m)$ depend on the current input $x_t$ and previous state $h_{t-1}$. The parameters $\phi$ and $\theta_m$ for $m = 1, \ldots, M$ are shared across all timesteps.

## B  Toy Regression Problem

To demonstrate the ability of our algorithm to learn conditional execution, we constructed the following regression problem: For each data point $(x_n, y_n)$, the input vectors $x_n$ are generated from a mixture of Gaussians with two components with uniform latent mixture probabilities $p(s_n = 1) = p(s_n = 2) = \frac{1}{2}$ according to $x_n|s_n \sim \mathcal{N}(x_n|\mu_{s_n}, \Sigma_{s_n})$. Depending on the component $s_n$, the target $y_n$ is generated by linearly transforming the input $x_n$ according to

$$y_n = \begin{cases} Rx_n & \text{if } s_n = 1 \\ Sx_n & \text{otherwise} \end{cases} \tag{1}$$

where $R$ is a randomly generated rotation matrix and $S$ is a diagonal matrix with random scaling factors. Figure 1 shows one possible such dataset.

Figure 1: The toy regression problem. Input data from a mixture of Gaussians (left) is transformed by scaling data from one mixture component and rotating the other around the origin yielding the target data (right).

## C  Language Modeling Experiments

We use a batch size of 128 elements and unroll the network for 35 steps, training with input sequences of that length. Each module has only 8 units. For the E-step, we sample $S = 10$ paths and run 15 gradient steps in each partial M-step.

# D  Image Classification Experiments

We use three modular layers with 3x3 convolutional kernels, striding two, and a single output channel followed by two fully connected modular layers with 48 hidden units and 10 output units respectively. We execute 2 out of 10 modules in each layer and concatenate their outputs. The baseline is non-modular with the same number of parameters that 2 modules would have. In effect, this roughly matches the amount of computation that is required to run the modular and non-modular variant.