[Reviews · NeurIPS 2018]

Reviewer 1



This work proposes a new way to learn modular networks, i.e., multiple networks for different tasks, and a controller that assigns tasks to networks. The authors propose a Viterbi-EM algorithm to learn the model parameters. The presentation is clear and the technical part seems to be correct. One consideration I has is the experiments. While modular networks should work best on very large datasets, this paper only presents experiments on small datasets (language modeling only has 0.9 million words, and image classification is just cifar10). There is no evidence on how the proposed model work on very large datasets.

Reviewer 2



The paper is concerned with conditional computation, which is an interesting topic yet at early stages of research, and as such one that requires much research and investigation. The paper proposes a latent-variable approach to constructing modular networks, modeling the choice of processing modules in a layer as a discrete latent variable. A modular network is composed of L modular layers, each comprised of M modules and a controller. Each module is a function (standard layer) f_i(x; \theta_i). The controller accepts the input, chooses K of the M modules to process the input, and outputs the as the module output. Modular layers can be stacked, or placed anywhere inside a standard network. The input/output of a modular layer is a tensor, and as such the layer is easily integrated with standard networks. Interestingly, the paper proposes a controller that outputs a distribution over compositions of modules (i.e., assigns a probability to each of the (M choose K) possible compositions). This is different from Shazeer et al., who assign a probability to each module and choose the K most likely. The key issue is a training/inference procedure, which is complicated due to the nature of the discrete latent variable. The authors propose trianing the model with Viterbi EM, optimizing a lower bound by introducing a q distribution over $a$, the choice of modules. The authors also mention REINFORCE and noisy top-K as options for training the model, though note that these require additional regularization in order to not collapse modules. Pros: - Key contribution: conditional computation layer that selects K of M modules per layer by assigning probability to the complete set (more general than greedy assignment as in Shazeer, et al., 2017), and training with EM explicitly leverages sparse computation (as opposed to MOE). - Simple, but powerful, modular layer for conditional computation that can be stacked and assembled in a straight-forward manner. - Well justified training procedure using EM to optimize a lower-bound over the marginal likelihood of y. - Experiments on Penn TreeBank and Cifar100 imply that the method is scalable to larger('ish) datasets. Cons: ----- - Contribution is somewhat incremental in that the the core idea was introduced by Shazeer et al. 2017 - Related work is somewhat sparse. - Biggest issue is the experimental results: the model does not work well on either benchmark at all: test accuracy was very low and perplexity very high compared to standard-compute models, though these are not shown in the paper. As the authors mention, there is yet a lot of work to make conditional-compute models viable. My overall impression is that this is a good paper that should be accepted at NIPS. The paper is clearly written, and easy to read. It links nicely to existing work in the field, clearly stating its contributions. The model design and derivation of the training procedure are clearly explained and justified. The paper makes a reasonable contribution in the area of conditional-compute by suggesting distributions over K modules per layer rather than softmax with top-K. This enables the use of a reasonable q distribution and generlized EM for training. Despite the weak experimental results, the paper represents a step in the direction for conditional neural computation.

Reviewer 3



This paper proposed training method for neural module network based on generalised EM algorithm. Based on the proposed method, the modules tends to be specialised and the modules tend to be selected deterministically. Additionally, the proposed training method eased the importance of regularisation for avoiding module collapse. Training modular neural architecture without additional data or complicated regularization strategy is an interesting and important problem. Many papers addressed this problem in the name of conditional computation and many training procedures and regularization method has been proposed. At the best of my knowledge, the training method proposed in this paper is not explored before and the experimental results suggests that the method might be an effective way to training such modules. However, I'm not sure whether the illustrated experimental result is convincing enough. Following lists are some of my concerns: - Reinforcement baselines Performance of reinforcement algorithms are sensitive to detailed implementation e.g. appropriate use of baseline for the log-likelihood value. While usage of baseline might make a huge difference in practice, it is not mentioned in the paper. - Continuous relaxation I think trivial baseline for learning module selection if using continuous relaxation of discrete random variable by using Gumbel softmax tricks. As this approach makes model differentiable at the initial stage of the training, this could be a strong baseline which should be compared to the proposed model. - Comparison with baseline for Language modeling (Table 1) Language modeling is a task that could be benefit from large number of parameters. Therefore, I believe we need to make sure whether the performance gain is comming from the module selection or simply from larger parameters. One way to confirm this would be running baseline with (M=5, K=5) or (M=15, K=15) and make comparison with EM Modular networks. - Stability of the random seeds I believe studying stability of the algorithm would be necessary to validate the algorithm and facilitate future use of the method. For example, sensitivity to random seeds, effect of using different number of modules and the number of samples for module selection variable could be studied. [After author response] I'm satisfied with the author response and most of my concerns are resolved. In that I increase my score to 7. I recommend authors to include discussions noted in the author response in the main paper if the paper is accepted.